# Bio-Inspired Textiles for Self-Driven Oil–Water Separation—A Simulative Analysis of Fluid Transport

**DOI:** 10.3390/biomimetics9050261

**Published:** 2024-04-25

**Authors:** Leonie Beek, Jan-Eric Skirde, Musa Akdere, Thomas Gries

**Affiliations:** Institut für Textiltechnik of RWTH Aachen University, Otto-Blumenthal-Straße 1, 52074 Aachen, Germany; jan.skirde@rwth-aachen.de (J.-E.S.); musa.akdere@ita.rwth-aachen.de (M.A.); thomas.gries@ita.rwth-aachen.de (T.G.)

**Keywords:** *Salvinia molesta*, bionic oil adsorber (BOA), oil–water separation, fluid transport, textiles, fluid simulation

## Abstract

In addition to water repellency, superhydrophobic leaves of plants such as *Salvinia molesta* adsorb oil and separate it from water surfaces. This phenomenon has been the inspiration for a new method of oil–water separation, the bionic oil adsorber (BOA). In this paper, we show how the biological effect can be abstracted and transferred to technical textiles, in this case knitted spacer textiles hydrophobized with a layered silicate, oriented at the biology push approach. Subsequently, the transport of the oil within the bio-inspired textile is analyzed by a three-dimensional fluid simulation. This fluid simulation shows that the textile can be optimized by reducing the pile yarn length, increasing the pile yarn spacing, and increasing the pile yarn diameter. For the first time, it has been possible with this simulation to optimize the bio-inspired textile with regard to oil transport with little effort and thus enable the successful implementation of a self-driven and sustainable oil removal method.

## 1. Introduction

Over the last 20 years, the production and consumption of oil, and hence the risk of oil pollution, have globally increased [1]. Despite the economically weakening effects of the COVID-19 pandemic, global oil production amounted to 4.4 billion tons in 2022 [2]. During the extraction, transportation, and usage of oil and oil-based products, accidents occur frequently, causing severe and sometimes irreversible pollution of the environment and harm to humans [3,4].

Various techniques exist for removing oil spills from water surfaces. Initially, the pollution is typically confined using oil booms to prevent further spread, and the contained oil can be incinerated in situ. In addition, dispersants can be utilized to break up the oil film into fine droplets, facilitating decomposition through bioremediation. Sorbents are another option, absorbing the oil and separating it from the water. Mechanical removal by using skimmers is also possible. Each method has inherent shortcomings that complicate their application, especially in the context of inland waters.

In the domain of technical applications, innovative solutions frequently emerge from the field of biology. The intricate surfaces of living organisms, shaped through millions of years of evolution, showcase optimized interactions with the environment [5,6]. Although these biological solutions may seem unconventional and challenging for material scientists to adapt, the examination of different species revealed a vast array of structures and functionalities [7]. Some species, in particular, exhibit outstanding water repellency and impressive oil adsorption properties [8,9]. Notably, the leaves of the floating fern *Salvinia molesta* were found to adsorb and transport oil on their surfaces, with the ability to release the oil for collection [8].

Investigating these structured surfaces proves that the oil adsorption rate correlates to several factors, including oil viscosity, hierarchical micro-architecture (such as size, shape, and spacing of microstructures such as hair), and surface wetting behavior. Inspired by these observations, especially in Salvinia leaves, it has been the intent to transfer this effect to technical surfaces for efficient oil and water separation. This approach results in a non-toxic, user-friendly, and versatile process for sustainable oil spill cleanup without additional energy input or waste generation. The outcome of this effort is a novel physical method for removing oil films from water surfaces, realized through a device known as a bionic oil adsorber (BOA) [8,10]. The BOA is a floating oil collector whose main components are a vessel and a technical surface inspired by the phenomenon seen in Salvinia. The surface is capable of separating the oil from the water and transporting it from the water surface to an oil container (Figure 1). This can be emptied at certain time intervals.

The technical surface used in the BOA for oil–water separation and oil transport is particularly competitive with adsorption materials that are currently used to remove oil spills. The materials used for this purpose are polymers, metals, ceramics, and also natural fibers. These are processed in various ways to form a hierarchically structured surface. In particular, these are sponges, foams, nets, textiles, nanofibers, and nanoporous structures [11,12,13,14,15,16,17,18,19,20,21,22,23,24,25,26,27,28,29,30,31,32]. Most research is currently being carried out on sponges and foams. Compared to these adsorption materials, the principle of BOA is characterized above all by the fact that it can be reused several times, the oil is collected directly, and the adsorption quantity is not limited by the size of the adsorption material. The latter is due to the fact that the bio-inspired material serves as a transport medium and does not just store the removed oil [10].

In this paper, we detail how the biological phenomenon can be abstracted and applied to a technical textile using the “Biology-Push” method outlined in VDI 6220 [33]. This specialized textile effectively separates and transports oil from water surfaces. Furthermore, we illustrate how the oil transport within the textile can be modeled through a three-dimensional simulation. With the results of this simulation, we derive how the textile construction can be optimized with regard to the fastest transport.

## 2. Abstraction of Biological Role Model and Transfer to Technology

### 2.1. Biomimetic Method

The development presented here is based on superhydrophobic surfaces such as the leaves of the floating fern *Salvinia molesta*. The already-mentioned phenomenon of oil–water separation through adsorption is used as inspiration for the development of a technical textile. For this purpose, the biology push process from VDI guideline 6220 [33] is implemented. In the biology push approach, a phenomenon is observed and researched in biological studies that serves as a biological model for technical development. The underlying mechanism or principle is then analyzed and detached from the biological model in the abstraction step. In the next step, the technical feasibility is checked, and a prototype can be developed. At the end of the process, a new product is introduced to the market.

### 2.2. Results of Biomimetic Process

The process observed in biology can be explained by various physical effects at different hierarchical levels. Starting with the smallest hierarchical level, wax crystals are found on the surface of the trichomes of *Salvinia molesta* (Figure 1, magnification). This makes the surface chemistry hydrophobic and oleophilic at the same time. In combination with the macrostructuring of the surface by the trichomes, this leads to superhydrophobicity, which results in a water contact angle of over 160° in *Salvinia molesta*. The result is the so-called Cassie-Baxter case [34]. This superhydrophobic behavior can be observed not only in *Salvinia molesta* but also in other oil-adsorbing surfaces, such as the water lettuce *Pistia stratiores*. The influence on wetting causes water to be repelled from the leaf surfaces and oil to be attracted, which in turn leads to the separation of the two liquid phases. In order to achieve this function, a hydrophobic and simultaneously oleophilic surface chemistry is necessary.

However, the macro-structuring not only leads to superhydrophobicity, but the trichomes also create a network of spaces filled with air. The combination of the hydrophobicity with the spaces creates an air layer under water, which can be recognized by a silver shine. This is caused by a total reflection at the air–water interface. If the leaf is in contact with oil, the air is displaced by the oil, and the oil is transported into the spaces due to capillary forces. The trichomes thus lead to the important effect of oil transport through the formation of a capillary network. Consequently, the trichomes must be abstracted. A closer look at the oil transport reveals that it only takes place at the base of the trichomes. These have a nearly circular cross-section [8]. Consequently, the trichomes can be abstracted as columns that protrude orthogonally from a surface.

In *Salvinia molesta*, it can also be observed that the velocity of oil transport varies depending on the position of the leaf. Barthlott et al. explain this by the different distances and diameters of the trichomes at the different positions on the leaf. The fastest transport takes place in the leaf areas where the distance between the trichomes is small [8]. The narrower spacing leads to higher capillary forces and thus to faster oil transport [10].

The entire leaf structure consists mainly of non-hardened, living cells. This allows the leaf and the individual trichomes to move with water currents and other mechanical loads. As a result, the retained air layer is not destroyed [35]. This makes the entire structure resistant to external mechanical influences. Following this, the abstracted model should be made of a soft base material, which builds up a structural elastic body.

The trichomes of *Salvinia molesta* are divided into four different segments at the tips, which converge at the end in a single hydrophilic tip. This part of the biological model is neither relevant for the separation of oil and water nor for oil transport, as oil transport only takes place at the base of the trichomes, and this design is also not found in other oil-adsorbing biological surfaces [8]. Therefore, the characteristic egg-beater structure of *Salvinia molesta* is not considered in abstraction and transferred to the oil-adsorbing textile.

Overall, it can be deduced from the biological models that a biologically inspired surface should consist of orthogonally protruding, hydrophobic columns. These form a network of capillary channels, and the hydrophobicity and macrostructure create a superhydrophobic, oleophilic overall structure. The entire structure must be elastic in order to withstand mechanical effects such as fluid flows (Figure 1).

All aspects can be implemented in technical textiles. In principle, various textile structures can be used. In preliminary tests, knitted spacer textiles proved to be particularly advantageous, as rapid oil transport was observed. All aspects of the abstracted biological model, the hydrophobic and oleophilic surface chemistry, the macrostructuring, the interconnected spaces by columns, and the soft material, are therefore implemented using a hydrophobized knitted spacer textile. Polyethylene terephthalate (PET) is used as the base material. The columns are realized by the so-called pile yarn of the spacer fabric (Figure 1). The pile yarn keeps two knitted surfaces at a distance, resulting in a textile with a pronounced thickness. In this case, a textile several millimeters thick is sufficient for the transfer of the biological model due to the scaling factors to be taken into account. The pile yarn, in combination with the knitted surfaces, leads to a structuring of the surface on the macro and meso level. To achieve superhydrophobicity, the entire textile is coated with a layered silicate (Tegotop 210™, Evonik AG, Essen, Germany) in a dip-coating process. This is deposited on the filament surface in the form of platelets, which leads to the desired structuring and at the same time ensures hydrophobic surface chemistry (Figure 2).

With the textile produced in this way, it is possible to transfer all functional aspects of superhydrophobic surfaces for oil–water separation into technology. The textile can be integrated into the BOA and, in its current prototype state, enables the removal of 4 L of diesel within one hour (see Appendix A for the oil adsorption in the bioinspired textile) [10].

### 2.3. Discussion of the Biomimetic Process

In *Salvinia molesta* and other superhydrophobic plants, oil separation and transport have not evolved for survival but are rather a side effect of superhydrophobicity. Therefore, the presented textile is more bio-inspired than a biomimetic product. Nevertheless, the biology-push method and thus a bionic method were successfully applied.

The textile differs from various other bionic implementations of *Salvinia molesta* [36,37,38,39,40,41], as the aim was not exclusively to transfer the superhydrophobicity, but to implement an additional function, which is oil transport. For this reason, the distinctive eggbeater-like heads seen in *Salvinia molesta* were also not abstracted and transferred. They have no function in this context and are therefore not to be found in the technical implementation, unlike other technical surfaces inspired by *Salvinia molesta*.

Overall, the functions of oil–water separation and oil transport can be successfully implemented inside a textile. The textile can not only be used as an adsorption material on its own but can also be used to implement the bionic oil adsorber. It not only separates oil from water, but also transports the oil into a container so that it can be collected there directly. Due to gravity, the oil desorbs without additional stripping or similar measures [10].

Current studies show that the BOA and, in particular, the textile can be used continuously over a period of 30 days [9]. However, the oil must be removed from the textile at some point. Initial tests have shown that this is possible by spinning the textiles or rinsing them with solvents. Following this, they can theoretically be reused. The maximum reuse rate has not yet been determined. At the end of their life, the textiles will most likely have to be incinerated. Until then, they have the immense advantage over conventional adsorption materials that they continuously adsorb and transport oil over a very long period of time and can be reused after use.

## 3. Oil Transport Inside the Bio-Inspired Textile

Using the bionic process and screening various textiles, it was possible to find a textile that enables bio-inspired transport. This allows the principle of the bionic oil adsorber to be implemented [10]. The production of spacer textiles is a highly productive process. However, the set-up of the double Raschel warp-knitting machines used for this purpose is complex, which is why an experimental parameter variation to determine the optimum design parameters would be time-consuming and cost-intensive. The transport process has already been successfully understood using a two-dimensional simulation [42]. Nevertheless, a three-dimensional simulation is necessary to determine all parameters, which is why this will be carried out below on the basis of the bio-inspired textile presented.

### 3.1. Method

#### 3.1.1. Simulation Setup

In this work, the computational fluid dynamics (CFD) solver ANSYS^®^ Fluent, Ansys Inc., Canonsburg, PA, USA, is used for this purpose. To solve the Navier–Stokes equations (NSE, Equation (1)), the computational domain under consideration must be spatially discretized so that the partial differentials of the NSE can be converted into finite volumes. The resulting difference equations can be evaluated on a computational grid. To achieve this grid, the flow volume is filled with finite volume elements, whereby the volume elements can take the form of a hexahedron, a tetrahedron, or a prism. The NSEs are only solved at the nodal points. Discretization transforms the partial non-linear system of differential equations into an algebraic system of equations.
(1)$$\frac{\partial }{\partial t}\underset{V}{∭}\rho \vec{u}\;\mathrm{dV}+\iint_{A}\rho \vec{u}\vec{u}\cdot \vec{n}\;\mathrm{dA}=\;-\iint_{A}p\;\mathrm{dA}+\iint_{A}\bar{\bar{\tau }}\cdot \vec{n}\;\mathrm{dA}+\underset{V}{∭}\rho \vec{g}\;\mathrm{dV}$$

The CFD solver ANSYS^®^ Fluent uses the finite volume method for discretization. The fluid volume is first divided into a discrete number of control volumes by the ANSYS^®^ ICEM CFD (ICEM) grid generation software (version no. 14.0). As a result, the flow variables are stored on the grid points, which is why the control volume is not identical to the grid cell [43]. For each finite control volume, the NSE are formulated integrally, where $\vec{n}\;$ is the normal vector to the surfaces, and the volume integrals are converted into surface integrals using the Gaussian integral theorem.

The surface integrals in the NSEs are represented by the sum of the fluxes over the individual cell surfaces of the control volume. The spatial partial differentials in the convective and diffusive fluxes are approximated by differential expressions using Taylor series. Depending on the discretization, first- or second-order forward, backward, or central differences are applied. Depending on the order of the discretization of the differential equation, a truncation error arises. For an infinitesimally fine mesh, the discretization error is close to zero. This corresponds to the discretization step sizes approaching zero [43].

The partial time derivatives must also be approximated. Implicit time step methods are used for this purpose, as the maximum time step is not restricted by the Courant–Friedrichs–Lewy number (CFL number). The CFL number is a necessary condition to achieve stability with explicit time-stepping methods. For implicit methods, however, it can be neglected. If a scheme is consistent and stable, it is also convergent. As a result, the solution of the difference equation strives towards the solution of the partial differential equation [43,44,45].

Initial values and numerical and physical boundary conditions (bc) must be defined for the unambiguous solution of the NSE, as the transient NSE are parabolic-hyperbolic. The discretized NSEs can be solved with the initial and boundary conditions using various solution methods. These can be divided into centralized methods, upwind methods, and high-resolution methods [43]. Based on the conditions set up in this way, the system of equations consisting of the conservation equations, the thermodynamic equation of state, and the constitutive equation is closed and can therefore be solved. No further steps need to be applied for laminar flows.

During oil transport within the knitted spacer textile, oil and air interact with the solid surface, resulting in a multiphase flow. Each phase has homogeneous properties, whereby it is assumed here that the phases do not mix (separated model) [46,47]. The volume-of-fluid method (VoF) is the standard for simulating separated phases. It is also used in ANSYS^®^ Fluent. In this method, it is assumed that the volume fraction of a phase within a calculation cell is stored as a supplementary variable C. This is used to weight the flow variables and then calculate the share of the phase volume in the total cell volume [46,48]. The NSEs are then calculated for each cell and each phase, and a convective term is added (Equation (2)).
(2)$$\frac{\partial C}{\partial t}+\vec{u}\nabla \;C=0$$

It is relevant that the cell size of the calculation grid is smaller than the diameter of the individual fluid particles. The value of C is 1 if only phase 1 is present in the cell and 0 if only the other phase is present. The value is between 0 and 1 if both phases are present in a cell [46,48].

The knitted spacer textile presented in Section 2 is selected as the basis for modelling the bio-inspired textile. For modelling, it is necessary to obtain an image of the textile that is as accurate as possible, which is why three-dimensional images are generated by computer tomography (CT-ALPHA, ProCon X-ray GmbH, Sarstedt, Germany) in addition to the shown SEM images (Figure 3). Based on this, the pile yarn spacing (A = 0.8 mm), the pile yarn length (S = 2.1 mm), and the pile yarn diameter (D = 66 µm) are determined.

Based on this, a computer-aided design model (CAD model) is created in Inventor^®^ (Autodesk, Inc., San Rafael, CA, USA). It should be noted that the pile filaments are slightly curved. In the side view, the shape can be seen as an inverted C. In the front view, it resembles an S-shape. Four pile filaments converge at each node. The shape of the pile filament is fixed and is not influenced by the simulated flow. There are spaces between the pile filaments, which are called corridors. Due to the symmetry of the textile, the model is scaled down such that two to three nodes can be viewed. As a result, a total of three corridors are simulated. The rest of the textile is modelled using the boundary conditions in the CFD solver.

The CAD model is meshed using the ICEM mesher, with a total element number of 8.1 million. The grid is then imported into the CFD solver ANSYS^®^ Fluent, and the boundary conditions are defined. On the left is the inlet through which diesel flows into the air-filled fluid domain. This is realized by a pressure boundary condition. The textile is bounded at the top and bottom by a solid surface. An oil contact angle of 20° is present on this and the pile filament surface, whereby the surfaces are wettable and an adhesion condition is present. The boundary condition on the piles’ surface is set as a no-slip boundary condition A periodic boundary condition is provided at the sides so that fluid can flow in and out there. Air and diesel flow out at the outlet opposite the inlet. A pressure boundary condition is provided at the inlet and outlet, whereby the pressure difference is 0 Pa. This corresponds to the pressure difference that exists in the real implementation. An overview of the boundary conditions can be found in Figure 4a.

Up to now, the volume flow rate has been selected as the reference variable. In the simulation, however, the transport speed can also be evaluated directly and easily, which is also independent of the sample size. Therefore, in this case, the oil transport velocity is selected as the comparative variable, whereby this is averaged in terms of location and time. For the evaluation, vertical planes are inserted into the fluid domain at intervals of 0.25 mm. The planes intersect the pile filaments in different proportions in order to evaluate the influence of the pile filaments on the velocity (Figure 4b). On each plane, the transport velocity is area-weighted and averaged by projecting the adjacent cells onto the plane. At a fixed point in time, the velocity on the plane is evaluated, whereby smaller cells are weighted less than larger cells. This is to avoid an overestimation of the areas around the pile filaments and the nodal points due to the locally finer grid. The averaged velocities per plane are summarized as a total oil transport velocity. A preliminary study has shown that the oil transport speed fluctuates per time step, resulting in fluctuating curves over time. For this reason, the speed is not only averaged locally but also arithmetically over time. The resulting value is then referred to as the oil transport velocity.

#### 3.1.2. Method of Experimental Validation of Simulation

To verify the simulation results, the transport process in the cross-section of the knitted spacer fabric must be examined in detail. For this purpose, the knitted spacer fabric, which served as the starting point for the simulation, is examined more closely with diesel fuel. The textile is an industrially produced knitted spacer fabric taken from Müller Textile GmbH, Wiehl-Drabenderhöhe, Germany. It is made from polyethylenterephthalat (PET). The material itself shows a water contact angle of ~70° and the whole textile structure of ~150°. Diesel is a conventional diesel fuel that was procured from Aral AG, Bochum, Germany. This diesel fuel has a density of 0.845 g/cm³ and a viscosity of 4.5 cP. Diesel fuel is a suitable test medium as it is easy to obtain, has a low viscosity, the properties can be easily implemented in the simulation, and the properties are highly consistent, allowing for reproducible test conditions. In addition, it is highly relevant in terms of clean-up methods for oil spills, as most of the oil to be removed, an estimated 1,200,399 MT, is spilled during onshore oil use [49]. On land, diesel is often used as a fuel for motor vehicles and large engines.

From the textile, 20 samples measuring 10 mm × 60 mm are cut in the direction of production. The sample is mounted by means of laboratory clamps in order to come into contact with oil. The horizontal transport distance is recorded at intervals of 30 s over 11 min with a camera from Fujifilm Holdings K.K., Tokyo, Japan, and the images are analysed with the program GNU Image Manipulation Program (GIMP). The experiments are only carried out horizontally. The oil transport velocity is calculated from the series of images based on the flow edge and corresponding transport height. Additionally, the transport is evaluated qualitatively. In the course of this work, the oil volume flow is not quantified, as this cannot be compared with the simulation carried out. In previous studies, a maximum volume flow of 500 mL/h with diesel was determined for the spacer textile used.

#### 3.1.3. Method of Parameter Variation

The preceding analysis of the flow field makes it possible to carry out a study to determine the influence of the textile design parameters. The aim is to determine the influences of the pile thread diameter, the fiber spacing, and the pile length on the oil transport speed.

First, a statistical test plan is created as a 2^n^-factor test plan. The plan is based on a slightly adapted version of the bio-inspired textile presented in Figure 2. The pile yarn diameter D is 70 µm, the pile yarn spacing A is 0.8 mm, and the pile length S is set at 2.1 mm. A variation of ±50% is used for the variation in the test plan (Table 1, Appendix B, Figure A1).

When designing the changes in the CAD model, a change in the curvature of the pile yarn results in a change in the pile length and the yarn spacing. Consequently, a shorter pile length leads to a stronger curvature, and vice versa for a longer pile length (Figure A1).

When meshing the CAD models, the periodic boundary condition at the upper and lower boundaries can no longer be used, as otherwise the cells would not meet the quality criteria in ANSYS^®^ Fluent. As a result, the calculation would no longer converge. Therefore, the periodic boundary condition is removed. In order to determine the influence of this change, a simulation with and without periodic boundary conditions is carried out on the basic settings. In the latter case, the two side surfaces are replaced by frictionless walls with a contact angle of 90°, whereby no adhesion condition is generated and no wetting takes place that could distort the transport speed.

The oil transport velocity is 3.02 mm/s in the calculation with the periodic boundary condition and 2.84 mm/s without. The deviation is 6%. The flow fields also appear qualitatively very similar. The effect can therefore be rated as small. Despite this negligence, the grids must still be smoothed in order to achieve sufficient mesh quality. A maximum violation of the geometry of 10% is permitted. This corresponds to a maximum displacement of the individual elements of 0.01 mm.

The calculation and evaluation of the oil transport in the parameter variants is carried out in the same way as the method described in Section 3.1.1.

Based on the findings for the parameter variation, two further textile variants are modelled in CAD with values changed by 50% and 25% compared to the best variant from the parameter study. These have a reduced pile length of 0.5 mm and 0.75 mm, respectively; the yarn spacing is increased to 1.8 mm and 1.6 mm; and the pile yarn diameter is increased to 210 µm and 157.5 µm. Both variants are as evaluated as the other variants and are named i and j.

### 3.2. Results of Investigation of Oil-Transport inside the Bio-Inspired Textile

#### 3.2.1. Analysis of the Simulated Flow Field

The evaluated transport velocity of the oil in the textile fluctuates over the simulation period. At the start of the simulation, the speed is at its highest at approx. 6 mm/s and drops to a minimum of approx. 2.2 mm/s in between (Figure 5). The total duration of the simulation is 0.8 s. After that, the oil flowed through the simulated area. On average, the transport speed is therefore approx. 3 mm/s.

The flow edge of the oil is curved concavely towards the upper and lower limits of the simulated textile, resulting in a C-like structure. In addition, a curvature of the flow edge can be seen on the surface of the pile yarns. The flow edge can be recognized as a clearly coherent boundary between oil and water, as the homogeneous model was selected in the simulation. The curvature of the flow edge changes with increasing distance from the pile yarns: the greater the distance, the less curved it is (Figure 6, t = 0.25 s).

If the flow direction is represented by length-normalized vectors on the flow edge, it can be seen that these always point to the nearest wettable surface. This results in the observed C-shape (Figure 7).

In addition to the flow edge, the velocity distribution parallel and orthogonal to the flow direction is investigated in order to evaluate the areas in which the oil flows particularly fast. Here, the velocity is limited to 0 to 6 mm/s, and the vectors are normalized and projected tangentially into the plane. This only allows a statement to be made about the direction, not about the magnitude of the vectors. The adhesion condition at the upper and lower boundaries and at all other walls can be recognized by a velocity of 0 mm/s. The maximum velocities (Figure 8, red) are between the pile yarns. In the direction of flow behind the pile yarns, areas with reduced velocity can be observed. This results in the formation of regular flow channels between the pole yarns, in which the majority of the oil transport takes place in directed paths.

#### 3.2.2. Experimental Validation of Simulation

The transport velocity in the experiments is approx. 0.7 mm/s. If this is compared with the minimum and maximum speeds of 2.2 mm/s and 6 mm/s calculated in the simulation, there is a deviation of 68 to 88%. If the discretization error of 61 to 80% is taken into account, this results in a velocity range of 0.6 to 1.2 mm/s. The deviation between the velocity determined in the simulation and the measured velocity is therefore of the same order of magnitude. As a result, it was possible to confirm the absolute oil transport speed in the simulations.

Besides the quantitative validation, the qualitative observation of the flow is relevant in order to prove the physical correctness. The C-shape of the flow edge, recognizable in the simulation, can also be seen in the lateral view of the textile. The oil is also transported through the flow edge first at the upper and lower boundaries, i.e., at the top surfaces of the spacer fabric, and then trailed in the middle. This results in the same flow behavior in the simulation as in the experiment. The flow field is therefore physically reasonable.

#### 3.2.3. Parameter Variation

The transport velocities determined for the variants were between 2.25 mm/s and 4.06 mm/s (Figure 9). The fastest oil transport was determined for the variant with the largest pile yarn diameter, the largest yarn spacing, and the shortest pile length (variant b).

Smaller yarn spacings lead to a higher oil transport speed. The smaller the pile length, the lower the transport velocity in the simulation (Figure 8). The difference between the respective test points with large and small pile lengths is 13 and 18%, which is why the influence appears to be less than the influence of the filament spacing. The increase in the pile filament diameter leads to an increase in the oil transport velocity (Figure 9). The magnitude of the change varies by 8% (difference between variants g and h) and 48% (difference between variants d and c). Consequently, the influence of the pile yarn diameter is stronger with smaller pile lengths and smaller yarn spacings than with the higher versions of the two remaining parameters.

From the simulation, it can be seen that the curvature of the pile yarn also has an influence on the oil transport speed. The stronger the curvature is, the lower the oil transport velocity. From an analysis of the velocity vectors, it is visible that the flow direction is more directed to the upper and lower boundaries with a less pronounced curvature (Figure 10). The more curved the pile filaments are, the more strongly the vectors point in the predicted direction of flow.

For the two variants with optimized design parameters i and j, an oil transport speed of 4.28 mm/s, respectively, 4.55 mm/s is determined (Table 2). This corresponds to an improvement of 6%, respectively, 12%, in comparison to the best variant in the parameter variation (variant b).

### 3.3. Discussion

#### 3.3.1. General Oil Transport

The general flow field that can be seen in the simulation may be mainly explained by capillary forces acting inside the model. Due to capillary forces, the C-shape forms itself because the adhesive forces on the upper and lower boundaries are counteracting with the oil resulting from the surface tension [34,50,51]. As a consequence, the oil is pulled through the textile at the areas where the oil is in contact with solid surfaces. This is the case at the upper and lower boundary and on the yarn surfaces. The additional oil volume is drawn in by the internal cohesive forces [50]. The velocity vectors, which always point to the nearest solid surface, support this theory. This leads to the conclusion that an optimization of the transport is possible through an optimization of the arrangement and design of the pile yarn.

The transport velocity fluctuates as the adhesive forces within the textile vary in intensity. There is a high velocity at points where the flow edge forms a large contact area with the surfaces. This is the case where there is contact with the upper and lower boundaries and the pile yarns. Due to the contact, the resulting adhesive force is greater in the direction of flow, which increases the velocity. Consequently, the velocity is lower at points with a smaller contact surface. The difference between the local extremes decreases over the course of the simulation, as there is more and more oil volume in the system under consideration and the inertia increases as a result. The velocity fluctuation is thus dampened over time.

#### 3.3.2. Experimental Validation

It was possible to demonstrate that the oil transport velocity in the simulation is in the same range as it was measured in the textile. Nevertheless, the simulated velocity and the velocity measured in the laboratory are slightly different. This can be explained by various factors.

One reason is the modelling error in the CAD model. The textile geometry was significantly simplified, e.g., on the cover surfaces. In addition, the use of the homogeneous model results in an error that could be reduced by using the Euler model (disperse flow). However, this model is significantly more complex to use.

In addition, the contact angle between oil and textile is approximated at 20°, and only every 25th time step is considered in the evaluation. This results in a resolution error. Furthermore, the model is optimally aligned horizontally. This is not the case in the experiment.

Other oils can also be absorbed by the textile. These are heating oil, waste oil, bilge oil, and engine oil. In this work, an analysis with these oils was not carried out, as previous simulative and experimental studies have already shown that the oil transport velocity of other oils essentially changes with the increasing viscosity of the oils [9,10,42]. Overall, the simulated velocity field can be successfully validated with regard to the quantitative absolute velocity and the qualitative course. The order of magnitude of the velocity deviation is similar in simulation and experiment. In addition, the shapes of the flow edge in simulation and experiment correspond to each other. It is therefore possible to derive conclusions from the model about the real transport process.

#### 3.3.3. Parameter Variation

It was seen that smaller yarn spacings lead to higher transport velocities. This can be physically explained by the fact that the distance between the surfaces decreases with smaller filament spacing, and the capillary effect increases as a result. The capillary forces are also greater in relation to the otherwise acting forces due to the reduced fluid domain. This results in a higher oil transport speed. However, the distance between the yarns shall not be reduced too much, as otherwise the frictional forces on the surfaces will predominate and the transport velocity will be restricted.

In contrast to that, a smaller pile length reduces the oil transport velocity. This correlation does not match expectations, as the shorter pile length leads to smaller capillaries, and therefore the transport velocity should increase. One explanation for this could be the flow channels introduced in Section 3.2.1: Oil transport primarily takes place in these, but the shorter pile length leads to greater curvature of the pile yarns. The curvature reduces the flow channels.

The increase in pile yarn diameter is followed by a higher oil transport velocity. This can be explained by the stronger influence of the capillary force with smaller pile lengths and thread spacings.

To increase the oil transport velocity, the filament spacing and the pile filament diameter should be maximized based on this parameter study. The pile length should also be reduced, although there is a limit value here due to the relative increase in frictional forces in relation to the capillary forces.

As a last factor, it was evaluated that the curvature of the pile yarn has an influence on the transport velocity. A stronger curvature causes the capillary forces to act more towards the upper and lower boundaries than in the intended direction of flow. This results in a lower flow velocity. This explanation can be supported by looking at the direction of the velocity vectors in the side view of the simulated area. The change in the direction of the resulting capillary force can be seen here.

Following the optimized variants, it is recommended to design a knitted spacer textile with a pile length of 0.75 mm, a pile yarn of 1.6 mm, and a pile yarn diameter of 157.5 µm.

The simulation results can be used to further optimize the textile with regard to oil transport. Measures can also be taken to improve oil–water separation. For this purpose, not only the resistance of the hydrophobic coating but also the textile construction, such as the base material, the yarn, and the pattern notation, can be adapted.

## 4. Conclusions

Based on superhydrophobic biological surfaces, such as the leaves of the floating fern *Salvinia molesta*, a technical textile for oil–water separation can be derived by applying the VDI guideline 6220. Implementation is possible, in particular with knitted spacer textiles. In order to better understand the transport process of the oil in the textile and to derive the optimum design parameters for the textile, a 3D simulation of the transport process can be performed. The knitted spacer textile with monofilaments as pile yarn can be transferred to a CAD model by making small adjustments. The oil transport in the horizontal direction can be successfully modelled using a three-dimensional simulation in ANSYS^®^ Fluent and thus by applying the Navier–Stokes equations. The qualitative profile of the transport in the simulation corresponds to the profile in the corresponding real test. It is therefore a laminar flow. The transport velocities in the simulation and the laboratory experiment are in the same order of magnitude, whereas the simulation can therefore be successfully validated. The parameter study shows that the oil transport velocity can be increased by the following parameters:increasing the pile filament diameter;increasing the filament spacing;reducing the pile length up to a limit value.

In the simulation, the oil transport velocity can be increased to 4.55 mm/s by reducing the pile length to 0.75 mm, increasing the pile filament diameter to 157.5 µm, and increasing the fiber spacing to 1.6 mm.

Ideally, the pile yarns should be arranged in such a way that the oil can flow through the textile in so-called flow channels. This reduces disturbances in the flow due to frictional forces. The simulation can be used to model oil transport. This means that the oil transport observed in biology can be successfully transferred to a textile, and the design parameters of the textile can be optimized. As a result, the biologically inspired product can be optimized and launched on the market.

Only with the textile optimized in this way is the market-ready development of the BOA as a self-driven and sustainable method for oil removal possible. This makes it possible to implement the Salvinia effect in a completely new application, as previous implementations have essentially been limited to air retention [36,37,38,39,40,41]. Furthermore, in this paper, we can demonstrate how liquid transport in spacer textiles can be modelled, calculated, and analyzed. Therefore, the simulation used here can also provide highly relevant findings for, e.g., water-conducting textiles used for the irrigation of agricultural products.

## Figures and Tables

**Figure 1 biomimetics-09-00261-f001:**
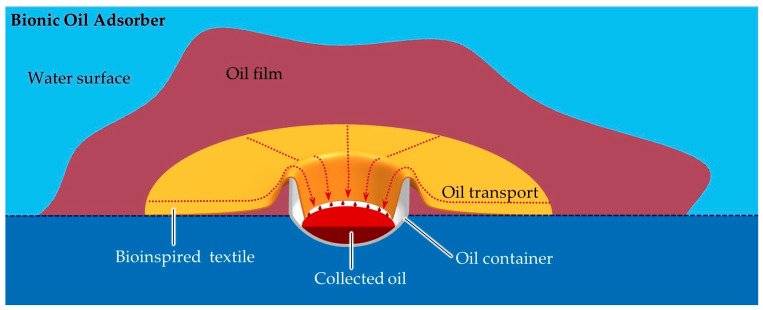
Cross-section of computer-aided (CAD) model of the bionic oil adsorber. The scheme shows an oil film (red) on a water surface (light blue). In the floating container (gray), the textile (orange) is fixed so that it is in contact with the oil film and the end protrudes into the container. The oil is adsorbed and transported by the BOA textile (red arrows). As shown in the cross-section, it enters the container, where it is released again and accumulates at the bottom of the container [10].

**Figure 2 biomimetics-09-00261-f002:**
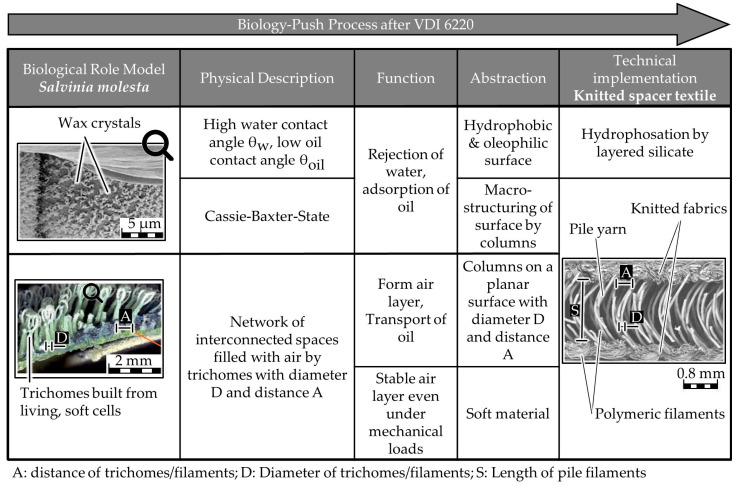
Biological model, physical effects and their impact with corresponding abstractions, microscopy images on the left above from [35], on the left below from [8].

**Figure 3 biomimetics-09-00261-f003:**
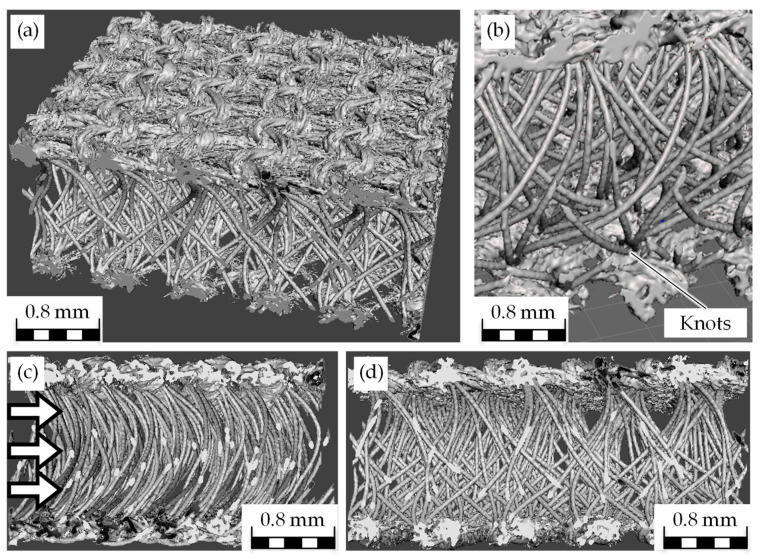
Computer tomography (CT) scans of the bio-inspired textile in isometric view (**a**), detailed view of the pile filament (**b**), inside view ((**c**), white arrows: inflow direction of the oil) and in front view (**d**).

**Figure 4 biomimetics-09-00261-f004:**
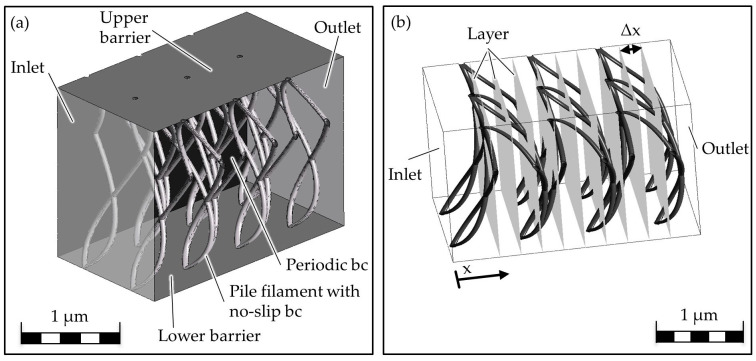
Schematic overview of the boundary conditions attached to the CAD model (**a**) and evaluation levels for determining the oil transport velocity (**b**); bc = boundary condition.

**Figure 5 biomimetics-09-00261-f005:**
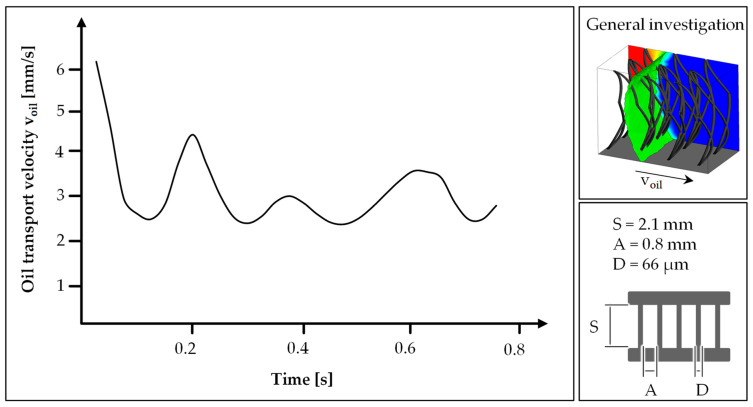
Oil transport velocity over time for the general simulation model with the textile design parameters taken from the µCT scan of the knitted spacer textile.

**Figure 6 biomimetics-09-00261-f006:**
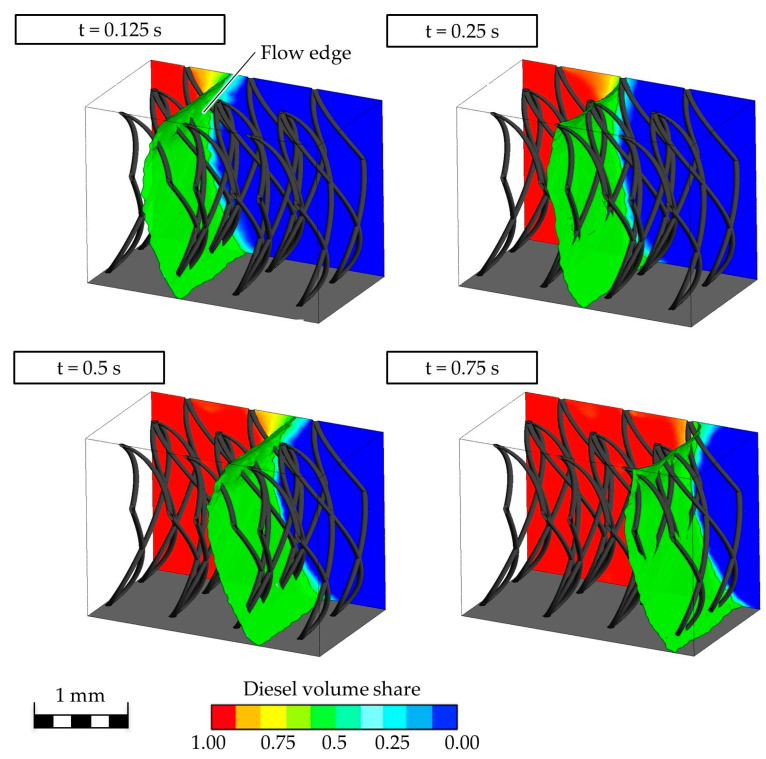
Flow edge of diesel fuel in the simulated model over time.

**Figure 7 biomimetics-09-00261-f007:**
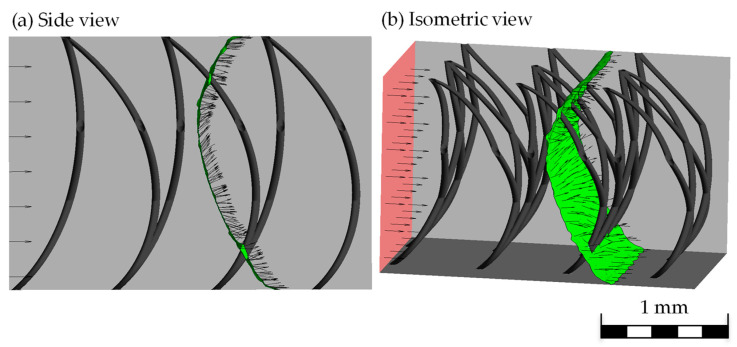
Velocity vectors (arrows) on the inlet (red) and the flow edge (green) inside view (**a**) and in isometric view (**b**).

**Figure 8 biomimetics-09-00261-f008:**
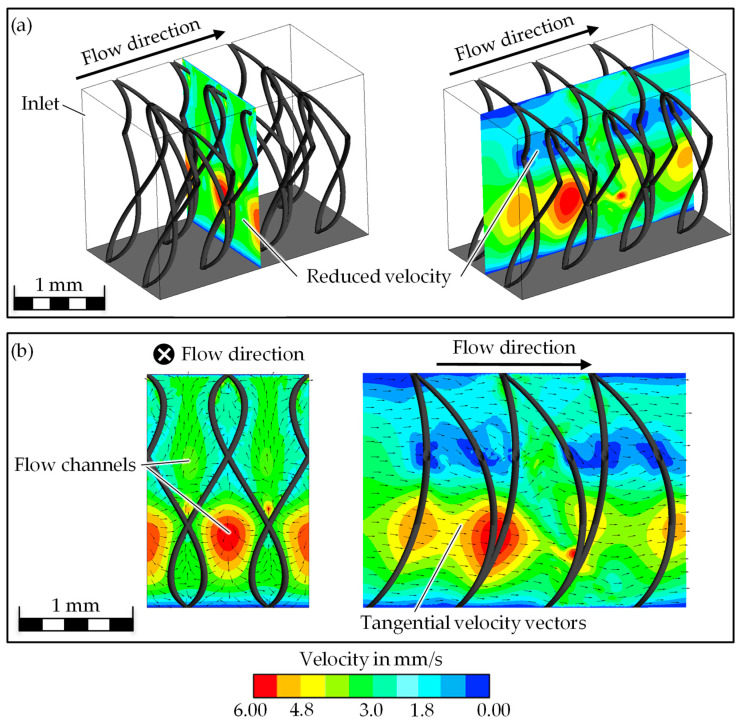
Distribution of the oil transport velocity on two planes in the fluid in isometric view (**a**) and side view (**b**), **left**: plane normal to the flow, **right**: plane tangential to the flow.

**Figure 9 biomimetics-09-00261-f009:**
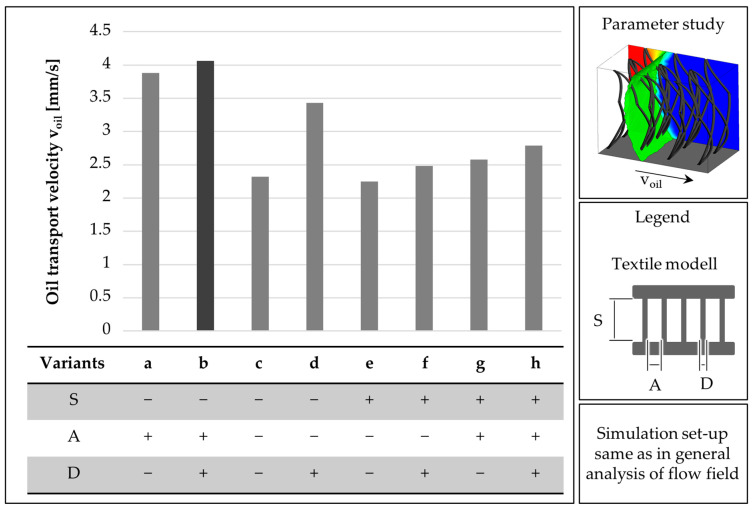
Oil transport speed of the variants of the parameter study, see Table 1 for details regarding the exact settings of + and − in the parameter study.

**Figure 10 biomimetics-09-00261-f010:**
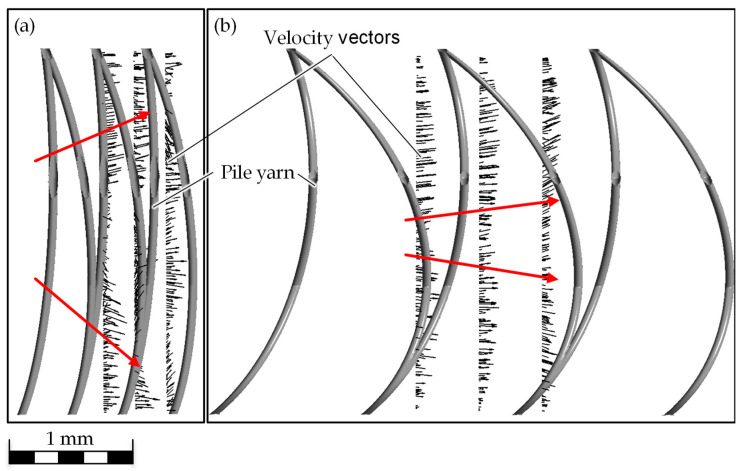
Visualization of the vectors of the oil transport velocity (arrows) in the side view for the smallest (**a**) and biggest (**b**) pile yarn distance.

**Table 1 biomimetics-09-00261-t001:** Overview of the configuration for the parameter variation.

Variant	Pile Length S [mm]		Pile Yarn Spacing A [mm]		Pile Yarn Diameter D [µm]	
a	−	1.05	+	1.2	−	35
b	−	1.05	+	1.2	+	105
c	−	1.05	−	0.4	−	35
d	−	1.05	−	0.4	+	105
e	+	3.15	−	0.4	−	35
f	+	3.15	−	0.4	+	105
g	+	3.15	+	1.2	−	35
h	+	3.15	+	1.2	+	105

**Table 2 biomimetics-09-00261-t002:** Overview over the design parameters and the oil transport velocity of the best variants simulated in the simulation.

Variant	Pile Length S [mm]	Pile Yarn Spacing A [mm]	Pile Yarn Diameter D [µm]	Oil Transport Velocity v_oil_ [mm/s]
b	1.05	1.2	105	4.06
i	0.5	1.8	210	4.28
j	0.75	1.6	157.5	4.55

## Data Availability

The raw data supporting the conclusions of this article will be made available by the authors on request.

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
