# Peer review of "Bio-Inspired Textiles for Self-Driven Oil–Water Separation—A Simulative Analysis of Fluid Transport"

_biomimetics, 2024, doi:10.3390/biomimetics9050261_

Round 1
Reviewer 1 Report
Comments and Suggestions for Authors
- I strongly recommend the addition of the list of symbols
- The desription of the numerical model is somehow messy.
- line 150: "the NSE can be converted into finite difference" - however further the Authors claim that they use finite volume method, not finite diference one
- Fig. 3a - it is not clear if on the wall on left side the periodic BC are fulfilled or it is the inlet
- How are defined the boundary conditions at the surfaces of piles (no-slip bc?)
- Does the shape of piles is given as steady and fixed - or it is a result of interaction with fluid flow? In the last case - how is the force calculated?
All these issues should be addressed.
- Line 251 "The transport is recorded..." - what EXACTLY is recorded? The velocity flow? If so - how was the velocity measured?
- line 475-480: "The parameter study shows that the oil transport velocity can be increased by (...) can be increased" - the repeated fragment. After that, the list of factors (lines 477-479) is somehow strange - I recommend to use the typical form with e.g. bullets
Reviewer 2 Report
Comments and Suggestions for Authors
Simulative analysis in this work has certain scientific and practical significance, but it is still far from sufficient to reflect its guidance for practical innovative products, Bio-inspired textiles. Firstly, the design parameters based on biomimetic models need to be analyzed and fully obtained. This is only mentioned in Figure 1 and its discussion. More importantly, the model design is completed without any actual corresponding biomimetic "products" (textiles) to verify it, resulting in a significant loss of credibility of the design.
Reviewer 3 Report
Comments and Suggestions for Authors
Comments on the Quality of English LanguageMinor editing of English language required
Reviewer 4 Report
Comments and Suggestions for Authors
Comments on the Quality of English Languageincluded in file
Round 2
Reviewer 2 Report
Comments and Suggestions for Authors
Good, no any more comments.
Reviewer 3 Report
Comments and Suggestions for Authors
The manuscript has been sufficiently improved to warrant publication in Biomimetics